

# A comprehensive review of sensor node deployment strategies for maximized coverage and energy efficiency in wireless sensor networks

Anusuya P.[1], Vanitha C. N.[1], Jaehyuk Cho[2] and Sathishkumar Veerappampalayam Easwaramoorthy[3]

[1] Department of Computer Science and Design, Kongu Engineering College, Erode, Tamil Nadu, India
[2] Department of Software Engineering, Jeonbuk National University, Jeonju, Republic of Korea
[3] Department of Computing and Information Systems, Sunway University, Malaysia, Kuala Lumpur, Malaysia

## ABSTRACT

Wireless Sensor Networks (WSNs) have paved the way for a wide array of applications, forming the backbone of systems like smart cities. These systems support various functions, including healthcare, environmental monitoring, traffic management, and infrastructure monitoring. WSNs consist of multiple interconnected sensor nodes and a base station, creating a network whose performance is heavily influenced by the placement of sensor nodes. Proper deployment is crucial as it maximizes coverage and minimizes unnecessary energy consumption. Ensuring effective sensor node deployment for optimal coverage and energy efficiency remains a significant research gap in WSNs. This review article focuses on optimization strategies for WSN deployment, addressing key research questions related to coverage maximization and energy-efficient algorithms. A common limitation of existing single-objective algorithms is their focus on optimizing either coverage or energy efficiency, but not both. To address this, the article explores a dual-objective optimization approach, formulated as maximizing coverage $\text{Max} \sum_{(i=1)}^{N} C_i$ and minimizing energy consumption $\text{Min} \sum_{(i=1)}^{N} E_i$ for the sensor nodes, to balance both objectives. The review analyses recent algorithms for WSN deployment, evaluates their performance, and provides a comprehensive comparative analysis, offering directions for future research and making a unique contribution to the literature.

## INTRODUCTION

Wireless Sensor Networks (WSNs) play a vital role in various fields due to their capabilities in wireless data collection, processing, and transmission (*Carlos-Mancilla, López-Mellado & Siller, 2016*). WSN comprises interconnected sensors that wirelessly transmit data to a central system or other sensors. Their applications span environmental monitoring, industrial automation, healthcare, and more (*Kandris et al., 2020*). WSNs are categorized

Corresponding author
Jaehyuk Cho, chojh@jbnu.ac.kr

into two main groups: centralized and distributed. In a centralized WSN, the activities of all nodes are controlled by a single device or a central entity. This central device typically coordinates the functions of all nodes within the network. Centralized models offer advantages such as centralized control and management, simplifying network operation and maintenance. However, they are susceptible to single points of failure, and scalability challenges may arise as the network expands. In contrast, distributed WSNs lack centralized control; each node operates autonomously. Every node in a distributed WSN has its own operating system and decision-making mechanism. Distributed models provide greater flexibility and scalability compared to centralized WSNs. Nodes can adapt to changing network conditions independently, enhancing the overall robustness and efficiency of the network. The choice between these two models depends on the specific requirements and constraints of the applications.

WSNs have multiple crucial parts (*BenSaleh et al., 2020*; *Cecílio & Furtado, 2014*) that work in harmony toward sensing, processing, and communication. Sensor nodes are the very basic building blocks within WSNs. Usually, a sensor node comprises various components like sensors, processors, transceivers, power sources, memory, the operating system, and a base station. Sensors are physical devices that collect data concerning different environmental parameters, for instance, temperature, moisture content, light intensity, or pressure. A processor (microcontroller) is a device that controls how the nodes operate as well as processes information from the sensors executing applications functions that are specific to it. A transceiver is used for wireless communication, where data can be sent and received from other networks through this process only. Sensor nodes are battery-powered, or they use energy harvesting techniques that may involve solar panels and vibration harvesters as power sources. Information is stored in terms of sensor data, program instructions, and configuration details in memory. Operating systems are specially designed for resource-constrained environments to ensure efficient task management by such devices. The base station node or sink acts as an interface between WSN and external networks such as the internet. This collects data from sensor nodes, and if necessary, it processes it before forwarding.

Sensors come in various types, each suited to specific applications, and they can be grouped based on lifetime, cost, material, power consumption, accuracy, precision, and application requirements (*Kumar & Reddy, 2020*; *Manjakkal et al., 2021*). Different sensing technologies are employed for different applications (*Dowlatshahi, Rafsanjani & Gupta, 2021*). Some common types of sensors include vibration sensors, temperature sensors, sound monitoring sensors, water flow monitoring sensors, and many others. In designing WSNs, it is crucial to consider the roles of network devices, performance metrics, and the operating environment. This includes understanding factors such as node roles (*e.g.*, sensing nodes, relay nodes, sink nodes), communication protocols, data transmission rates, power consumption, and environmental conditions (*Borges, Velez & Lebres, 2014*). The design of WSNs involves both low-level and high-level approaches. Low-level design deals with the specifics of individual components and protocols within the network, addressing issues such as communication protocols, power management, and data aggregation. High-level design focuses on overall system architecture, scalability, and

optimization. High-level approaches often build upon and refine solutions developed at the low level, addressing broader system-level concerns (*BenSaleh et al., 2020*).

WSN integration with Internet of Things (IoT) technologies (*Maini & Rani, 2024*; *Sidhu, 2024*), enhances the capabilities, flexibility, and usability of WSNs by creating new opportunities for applications in multiple domains, including smart cities, industrial automation, environmental monitoring, healthcare, agriculture, and many others. IoT ensures the following advantages. First, IoT guarantees enhanced connectivity, as its communication protocols and standards allow sensor nodes to connect and other IoT devices. Thus, WSNs can capitalize on wider communication infrastructures, including Wi-Fi, Bluetooth, Zigbee, or cellular networks, to transmit data across larger distances to the Internet. Second, IoT promotes scalability, as IoT platforms and architectures are developed to maintain large-scale infrastructure of interconnected IoT devices. Consequently, WSNs can easily increase the size of the network by adding new sensor nodes or integrating additional sensors and gadgets (*Jamshed et al., 2022*).

Localization of WSN is needed in multiple environments to understand and learn. The WSN environment may be terrestrial WSN (*Nasri, Mnasri & Val, 2020*), underwater WSN (*Chaudhary et al., 2022*; *Gola et al., 2023*), underground WSN (*Tam et al., 2020*), and space WSN (*Abdulwahid & Mishra, 2022*). Each WSN uses different routing (*Inga, Inga & Ortega, 2021*) and localization (*Mani et al., 2023*) approaches. A smart city (*Kirimtat et al., 2020*) utilizes many kinds of technological resources to improve the quality of living of its residents. To develop smart cities various technologies are put into use, forming a big network (*Ghazal et al., 2023*; *Gracias et al., 2023*; *Ramírez-Moreno et al., 2021*). Smart cities utilize digital technology to improve resource efficiency, reduce pollution, and enhance various aspects of urban life. They aim to improve daily life and reduce costs through the use of technology and the IoT (*Srivastava et al., 2022*). The IoT is being used to enhance economic performance and generate innovative technological solutions (*Khan, Chaturvedi & Jaswal, 2022*). General architecture of smart cities using WSN for multiple applications is given in Fig. 1. Many applications (*Ramesh et al., 2020*) fall under smart cities such as smart buildings (*Alsafery, Rana & Perera, 2023*; *Li et al., 2023a*; *Yang et al., 2023*), water quality monitoring (*Imran et al., 2020*; *Ramírez & Aragón-Zavala, 2023*; *Shahra & Wu, 2023*), traffic light control (*Sheela et al., 2023*), smart transportation (*Oladimeji et al., 2023*), smart agriculture (*Kumar et al., 2012*; *Bhatia, Jaffery & Mehfuz, 2023*; *Koshariya et al., 2023*), energy and waste management (*Alzahrani, Chauhdary & Alshdadi, 2023*), and pollution monitoring (*Henna et al., 2023*). Environmental monitoring in smart cities forms connected streets, which help with smart lighting, equipment accessibility, environment stations, connected charging stations, park assistance, connected waste containers, video management, WiFi or LiFi connections, automatic watering, transport and mobility, and public toilets. Smart cities provide several opportunities for advancement in research, with major challenges to be addressed (*Arroub et al., 2016*).

IoT for smart cities provides access to all necessary information from the cloud (*Abreu et al., 2017*; *Syed et al., 2021*). It provides smarter cities around the world. The deployment of sensor nodes is the most important area in developing the sensor network. It can be done by considering parameters, such as whether they should provide a better coverage

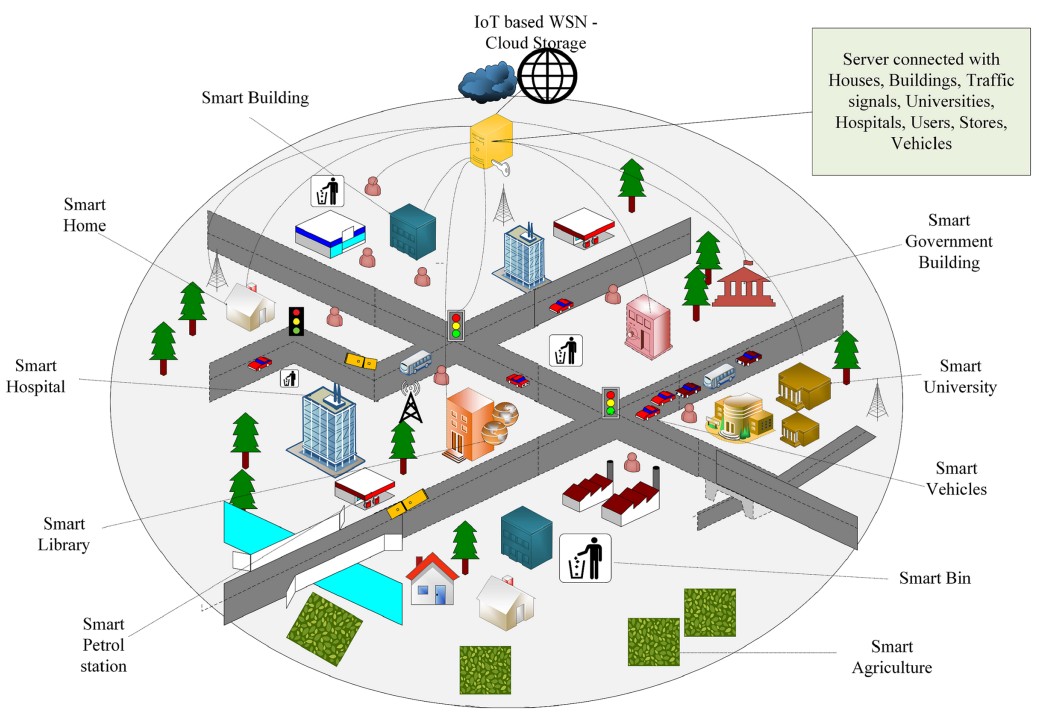

**Figure 1 WSN for smart city application.**

area and be connected to all networks. The power usage of the sensor nodes is also to be considered. The other consideration in deployment is the fault-tolerant (*Vihman, Kruusmaa & Raik, 2021*) capability of the nodes while sensing and transmitting. Security of data is one of the metrics considered in WSN. The process of positioning sensors in an appropriate location is known as "sensor deployment," which might be a challenge for researchers and programmers. In applications like forest areas, disaster management, and battlefields, sensor deployment is a difficult task; so use of an autonomous deployment technique by using airborne vehicles is proposed (*Sharma et al., 2023*). Deployment strategies (*Abdollahzadeh & Navimipour, 2016*; *Amutha, Sharma & Nagar, 2020*; *Priyadarshi, Gupta & Anurag, 2020a*) are the various techniques and protocols used to locate the sensors. Energy monitoring and savings depend on the routing protocols of the WSN (*Kumar & Reddy, 2020*). Technological advancements like artificial intelligence (*Lei, 2023*) and machine learning contribute to improving the metrics of performance of WSN (*Gillani, Niaz & Tayyab, 2021*; *Kim et al., 2020*). WSN uses machine learning (ML) for localization, routing protocols, MAC layer protocols, and energy-saving mechanisms (*Sharma, Haque & Blaabjerg, 2021*; *Vanitha et al., 2022*). Deep learning algorithm with neural networks becomes the emerging solution for optimization problem of the network (*Priyadarshi, 2024*; *Qiu, Ma & Priyadarshi, 2024*).

Research articles from platforms like Google Scholar and Web of Science provide valuable insights into the technologies used for deploying sensor nodes in WSNs. These technologies aim to meet parameters such as coverage and connectivity, power consumption, and fault tolerance. These technologies are continually evolving, with

ongoing research focusing on enhancing coverage, connectivity, power efficiency, and fault tolerance in WSN deployments. By leveraging insights from research articles, WSN designers and researchers can make informed decisions to optimize sensor node deployment strategies for various applications and scenarios.

Experimental performance evaluation of WSN (*Khalifeh et al., 2021*) can be conducted using various simulation tools and frameworks. Some of the commonly used tools and frameworks include (*Adday et al., 2024*; *Koyuncu & Bagwari, 2023*) network simulators, TOSSIM, MATLAB, and OMNET++. Several current WSN simulation frameworks (*Rajaram et al., 2016*) can also be integrated with real-time hardware prototypes, enabling researchers to validate simulation results in real-world scenarios. These frameworks facilitate the seamless transition from simulation to deployment, allowing for more accurate performance evaluation and validation of WSN designs. Furthermore, statistics obtained from government websites or other reliable sources can be used for real-time applications of WSNs. These statistics may include demographic data, environmental parameters, or other relevant information that can inform the design and deployment of WSNs in specific locations or applications. Many research articles propose real-time implementations of WSNs in various cities, leveraging simulation tools, frameworks, and real-world data to design and deploy sensor networks for applications such as environmental monitoring, traffic management, and urban planning (*Choudhary, Shrimali & Shreemali, 2023*). These implementations aim to address real-time challenges and improve the efficiency and effectiveness of urban infrastructure and services. The design and implementation of sensor nodes for specific applications should prioritize low cost, improved lifetime, and reduced energy consumption (*Khalifeh et al., 2021*; *Vanitha, Usha & Nanthiya, 2018*). The design and deployment of these nodes (*Kumar & Reddy, 2020*) should consider the application's requirements and the localization area of the nodes to enhance their effectiveness. Leveraging IoT sensors for interconnectivity *via* the internet and cloud storage access signifies a modern technological advancement, enabling seamless data exchange and storage. Furthermore, proper planning is essential in forming deployment strategies to ensure the optimal placement of sensor nodes in strategic locations, thereby maximizing their utility and efficiency within the network.

## SURVEY METHODOLOGY

Conducting a literature review on WSN deployment involves a series of systematic steps to ensure a thorough examination of existing research. This process includes formulating research questions based on the defined scope and objectives, selecting relevant databases, and utilizing appropriate search terms to identify relevant articles.

- This survey aims to explore WSN deployment strategies, focusing on key aspects such as node placement, coverage optimization, and energy efficiency to extend network lifetime. The primary goals are to identify current trends, challenges, and advancements in WSN deployment, assess various methodologies and algorithms, and pinpoint potential areas for future research.

Below are the formulated research questions used in this WSN research.

1) How can sensor nodes be deployed in WSNs to maximize coverage and minimize energy use?
2) What are the key considerations for deploying WSNs in various environments with diverse sensor types and communication protocols?
3) What are the recent advancements in deployment algorithms for WSNs, and how do they compare in terms of performance, especially regarding coverage and energy consumption?
4) What are the current trends and advancements in optimization algorithms for WSN deployment?
5) What future research areas can be explored based on the current state of WSN deployment strategies, and how can these address the current challenges?

- The key databases, such as IEEE Xplore, ACM Digital Library, SpringerLink, ScienceDirect, and Google Scholar, were used to gather relevant literature articles related to the research questions. This review focuses on articles published in the last 4 years (2020–2024) and some articles referred to before 2020 for coverage of basic concepts and recent advancements.
- The keywords used in the search of databases were "WSN," "WSN deployment," "sensor node placement," "coverage optimization," "energy efficiency in WSN," and "WSN in smart cities." The search was refined by using Boolean operators such as AND, OR, and NOT. The search terms were "WSN deployment for coverage AND energy efficiency," "Optimization algorithms for coverage OR connectivity," and so on.
- The included articles were peer-reviewed journal articles and conference articles, excluding the studies that do not directly relate to WSN deployment, *i.e.*, general WSN applications without a focus on deployment issues.
- The titles and abstracts of the identified articles were reviewed as an initial screening process, and then the full text of the article was analyzed depending on the search objective. The catalog was created by extracting the relevant data, including methodologies, algorithms, results, and conclusions, from the article.
- The articles were categorized based on the basic concepts of WSN, deployment strategies, coverage optimization techniques, and energy optimization techniques. Comparative analysis was conducted to find the similarities, differences, and research gaps in existing work, along with future research directions.
- Finally, the proposed article summarizes the key findings in optimization techniques for the deployment of WSN and gaps for future improvement through a literature survey.

The following procedure for the literature review is shown above in Fig. 2 as a step-by-step flow after selecting the study topics. A total of 148 of the 200 articles that were selected are being used as review articles. A review article was created when those article's concepts were examined and understood. Overview of this article is given in Fig. 3. The following sections in this article cover fundamental concepts commonly employed in WSN,

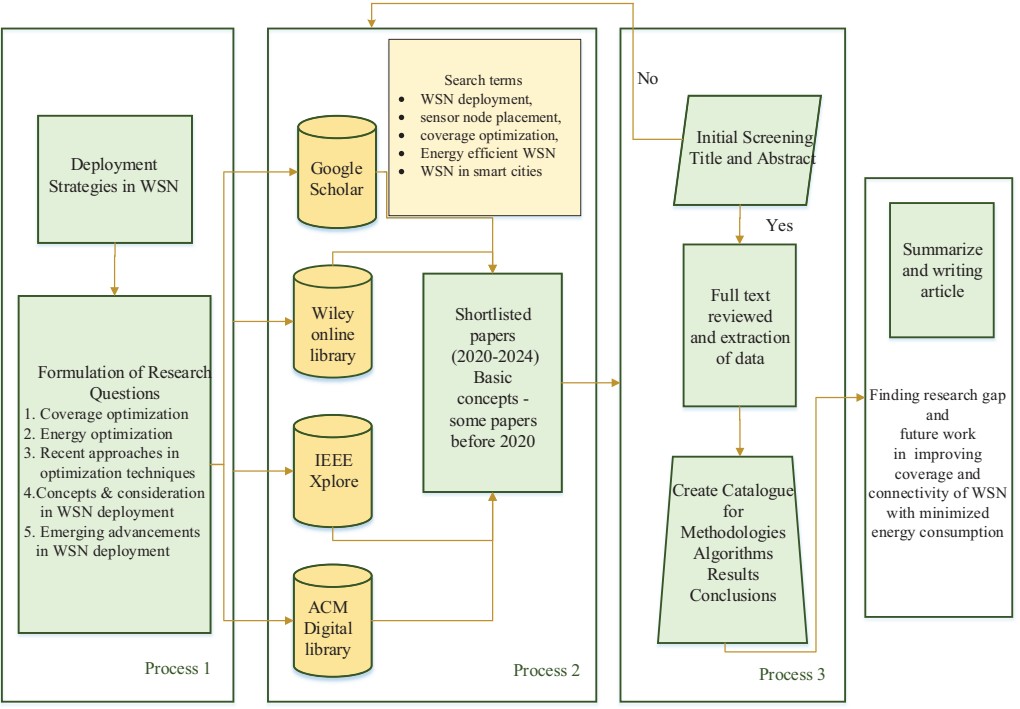

**Figure 2 Literature review methodology.**

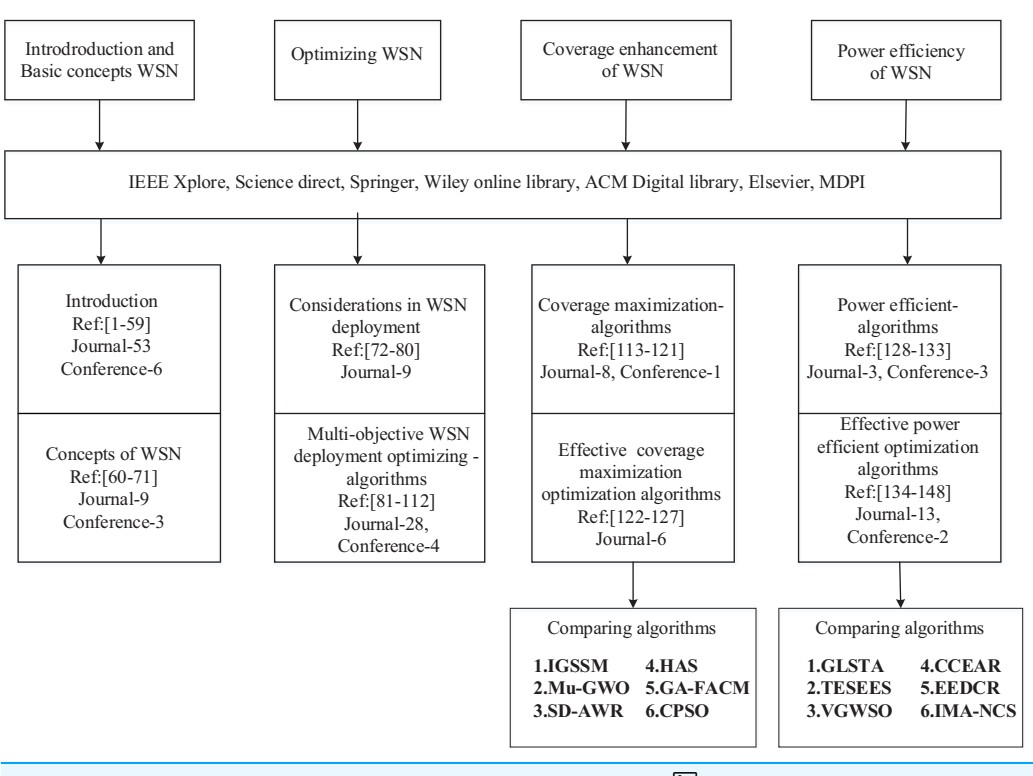

**Figure 3 Overview of review process.**

principles guiding deployment, newly suggested algorithms aiming to optimize sensor node deployment for enhanced coverage and energy efficiency, as well as an evaluation of these approaches including comparisons, encountered challenges, and concluding remarks drawn from the analysis conducted.

## GENERAL CONCEPTS

The deployment of sensor nodes in WSNs is crucial and can be approached using various concepts and strategies tailored to specific applications. Optimization algorithms, including heuristic techniques, are one such strategy utilized in node deployment to enhance network performance and efficiency (*Peyman et al., 2023*). General classification of WSN is explained below and referred to in Fig. 4.

### Classification of deployment method

Optimal placement of sensor nodes is essential for effectively gathering information about a specific area or region of interest (RoI) in a WSN. The type of deployment, the type of nodes, and the sensing model are determined (*Priyadarshi, Gupta & Anurag, 2020b*) by considering the characteristics of the RoI, WSN designers can strategically deploy sensor nodes to effectively monitor and gain information about the targeted area. Various references explaining different types of nodes, deployment methods, and sensing models are shown in Table 1.

#### *Random deployment and planned deployment*

Random deployment in WSNs is the placement of sensor nodes in an area without following any specific pattern or predetermined locations. Instead, sensor nodes are deployed randomly across the target area, often to achieve certain network performance objectives such as coverage, connectivity, and fault tolerance (*Dina, Deif & Gadallah, 2014*). Planned deployment in WSNs involves strategically placing sensor nodes in a predetermined manner based on specific criteria such as coverage, connectivity, energy efficiency, and application requirements. Unlike random deployment, which disperses nodes randomly across the area, planned deployment follows a structured approach to optimize the performance of the network (*Dina, Deif & Gadallah, 2014*).

#### *Heterogeneous nodes and Homogeneous nodes*

The performance of WSNs is influenced by the type of sensor nodes employed. Sensor nodes in WSNs can be grouped under two main categories based on their capabilities, functionalities, or characteristics. Firstly, heterogeneous nodes exhibit variability across multiple parameters, including processing power, communication range, energy storage, sensing capabilities, and deployment environments. In a heterogeneous cluster, diverse sensor nodes with varying abilities are utilized, enabling the network to cater to different requirements within the same cluster. This heterogeneity offers benefits such as enhanced scalability, flexibility, and robustness, facilitated by adaptive routing and task allocation strategies (*Wu & Chung, 2007*).

Conversely, homogeneous nodes are characterized by uniformity in their capabilities, functionalities, and characteristics within a cluster. These nodes possess identical storage

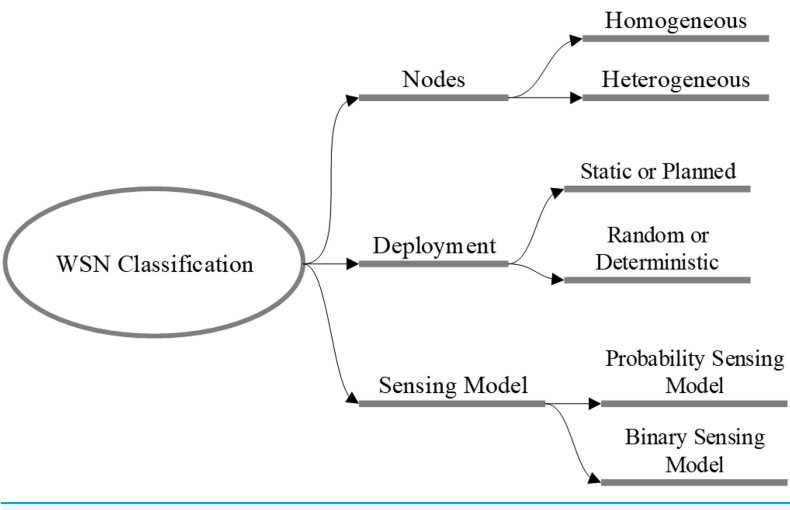

**Figure 4** **Classification of WSN.**               

**Table 1 Comparison of references based on the classification of WSN.**

| Reference | Deployment type | Sensor nodes | Sensing model |
|---|---|---|---|
| Dina, Deif & Gadallah (2014) | Random, Planned | Homogeneous, Heterogeneous | Binary, Probabilistic |
| Wu & Chung (2007) | Planned | Heterogeneous | Probabilistic |
| Singla & Kaur (2016) | Random | Homogeneous, Heterogeneous | Binary |
| Hossain, Biswas & Chakrabarti (2008) | Random, Planned | Homogeneous | Probabilistic |
| Soreanu & Volkovich (2009) | Random | Heterogeneous | Probabilistic |
| Altahir et al. (2021) | Planned | Homogeneous | Binary, Probabilistic |
| Akbarzadeh et al. (2012) | Random, Planned | Homogeneous, Heterogeneous | Probabilistic |
| Tripathi et al. (2018) | Random, Planned | Homogeneous, Heterogeneous | Binary, Probabilistic |

capacity and performance attributes, streamlining the deployment process and simplifying network management. The uniformity of homogeneous nodes results in consistent performance, reduced complexity, and cost-effectiveness, along with streamlined data processing (Singla & Kaur, 2016). The choice between heterogeneous and homogeneous sensor nodes in WSNs depends on the specific application requirements, with each approach offering distinct advantages in terms of flexibility, scalability, management, and performance.

### Binary sensing model and probabilistic sensing model

The sensing model encompasses various aspects, including the sensors' capabilities, the characteristics of the phenomena being sensed, and the methods used for data acquisition (Amutha, Sharma & Nagar, 2020). Proper deployment of sensor nodes depend on the sensing model that gives a better idea of how to locate the sensors by calculating their exact location (Ojeda et al., 2023). The two main classifications of sensing models were binary sensing model (BSM) and probabilistic sensing model (PSM) (Hossain, Biswas & Chakrabarti, 2008; Soreanu & Volkovich, 2009).

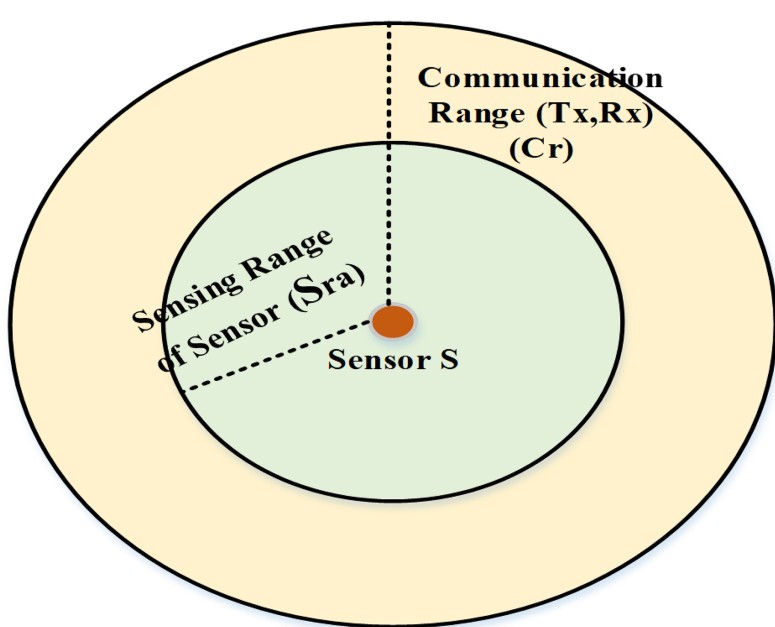

**Figure 5 Binary sensing model coverage.**     

A BSM is a simplification used by WSNs to explain how sensors sense events or happenings within their sensing scope. Within such a model, sensors decide whether an event of interest is present or not within their sensing area (*Altahir et al., 2021*). The diagrammatic representation of this model is given in Fig. 5. This model works on the assumption that sensors can only sense if the event is present or not but fail to give any information regarding its magnitude or intensity quantitatively. If an event e takes place within the location l the probability of sensing can be given by Eq. (1),

$$S_P = \begin{cases} 1, & if \ d(e,l) \leq S_r \\ 0, & otherwise \end{cases} \tag{1}$$

In a PSM, the sensing capacity of a sensor is characterized by a probability distribution rather than a binary decision. This means that instead of simply detecting the presence or absence of an event, the sensor provides a probabilistic assessment of the likelihood of the event being present within its coverage area. The probabilistic nature of the sensing model accounts for uncertainties and variations caused by external factors such as noise, interference, and environmental conditions (*Akbarzadeh et al., 2012*). Figure 6 represents the basic probabilistic model with varying sensing range. PSM for line of sight (LoS) coverage in sensor deployment was explained (*Akbarzadeh et al., 2012*). The general probabilistic coverage model (*Tripathi et al., 2018*) is given by Eq. (2), where the coverage probability at a location by sensor node S in a simple PSM model is,

$$P_c(l, S) = \begin{cases} e^{-\omega d(l, \varepsilon_i)}, & if \ d(l, \varepsilon_i) \leq S \\ 0, & otherwise \end{cases} \tag{2}$$

$\varepsilon_i$- Euclidean Distance, $\omega$- Characteristics of Sensor node S

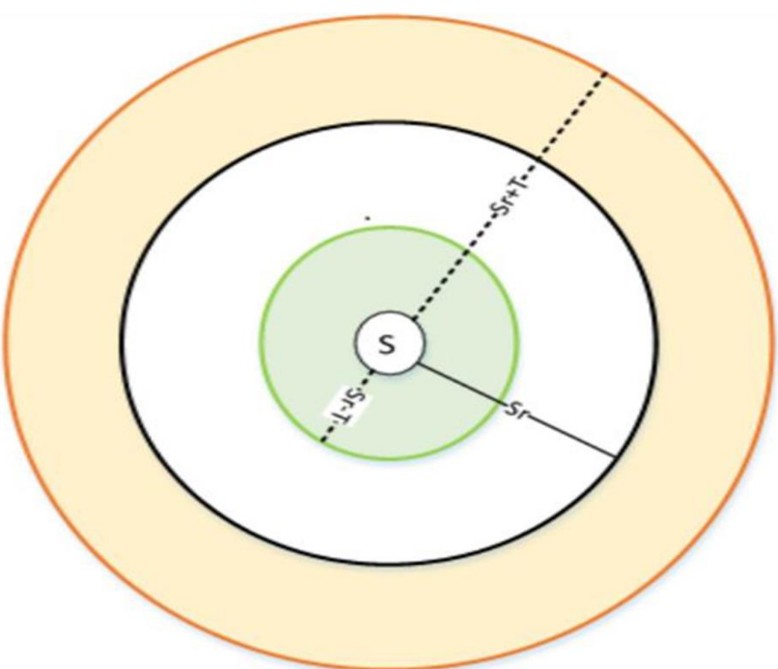

**Figure 6 Probabilistic sensing model coverage.**

Various probabilistic sensing models are found which are Log-normal shadowing, Elfes, and shadow fading models which are explained in *Altahir et al. (2021)*. These models are used depending on the target area covered by considering the field of interest (FoI).

### Diverse deployment environments

Establishing a WSN requires careful consideration of deployment areas. Indoor deployment, common for managing buildings and healthcare facilities, faces challenges such as signal interference (*Sharma et al., 2016*). Outdoor deployment covers large areas and involves coping with weather conditions and ensuring long-distance communication. Underwater WSNs face signal loss and corrosion challenges. Underground deployment, for mining and agriculture, requires signal transmission through soil or rock. Mobile WSNs, used in disaster response and wildlife tracking, face challenges in managing node movement and energy consumption. Sky-based deployment using drones offers extensive coverage but may have restricted carrying capacity and energy limitations.

## STRATEGIC CONSIDERATIONS IN WSN DEPLOYMENT

Strategically situating sensor nodes within a designated area is fundamental in WSN deployment to fulfill targeted goals such as environmental monitoring, data collection, or communication facilitation. Key considerations encompass coverage, power efficiency, fault tolerance, and security. The deployment process, illustrated in Fig. 7, underscores the importance of several factors to be considered during sensor node placement, elaborated as follows.

In a WSN, achieving coverage entails strategically placing sensor nodes to effectively monitor the designated zone or target area. Nodes must be positioned to ensure that every

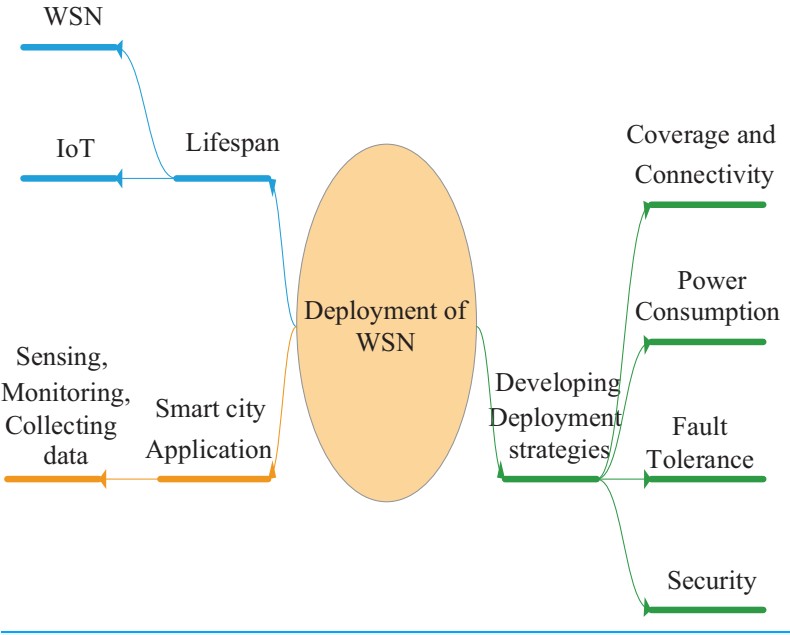

**Figure 7  Deployment optimization of WSN.**

part of the area falls within the sensing range of at least one sensor, maintaining complete coverage. The placement should also consider sustained connectivity between nodes (*Liu, 2015*). Coverage types vary based on the application's location: area coverage (or regional coverage) involves monitoring the entire region of interest, point coverage focuses on specific events or points, and barrier coverage monitors movement across protected areas or borders (*Tarnaris et al., 2020*).

Power management in WSNs is vital due to the limited power resources of sensor nodes, often powered by batteries or energy harvesters. Optimizing power usage is essential to ensure continuous network operation and prolong node lifespan. This optimization involves considering energy constraints and strategically placing nodes to conserve energy. Sensor node energy allocation (*Zhong & Wang, 2018*) for a complete cycle is given in Fig. 8. Maintaining low power consumption is crucial, particularly in applications like smart cities and military operations, where sensors must remain active. However, continuous activity can lead to increased power consumption and shortened node lifetimes.

Ensuring the resilience of sensor nodes is essential for maintaining network functionality in challenging conditions. Fault-tolerant deployment strategies are vital for reliable operation (*Gola, 2024*; *Lan, 2023*). Particularly in dynamic and harsh environments where nodes may encounter failures due to factors like energy depletion or hardware malfunctions. Common fault tolerance approaches include redundancy, error detection and correction, topology control, self-healing mechanisms, energy management, and resilient communication protocols. These mechanisms contribute to the reliability and robustness of WSNs, enabling continuous operation despite unreliable nodes and unpredictable environments.

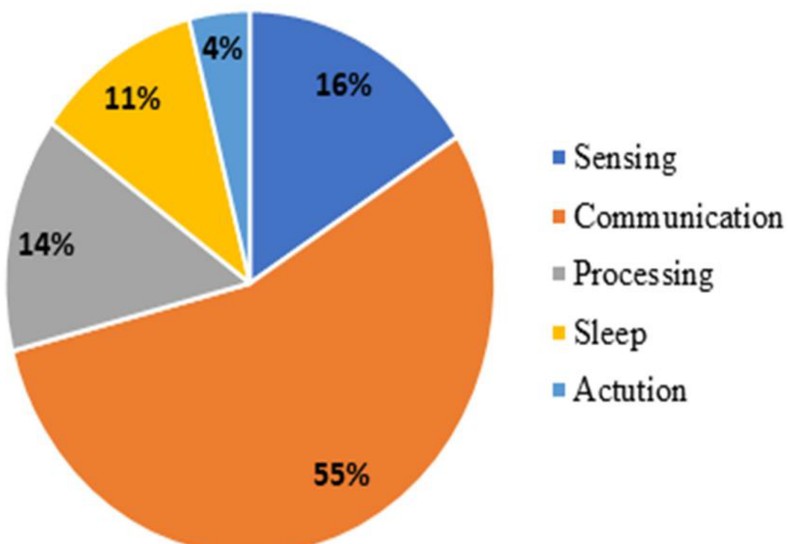

**Figure 8 Energy consumption for various functions of a sensor.**

Data security is essential in WSNs to prevent unauthorized access and modification. Security measures must be incorporated into the application to protect the data transmitted by the sensor nodes (*Adu-Manu et al., 2022*; *Faris et al., 2023*). Security mechanisms protect sensitive data, ensure the integrity of communications, and prevent unauthorized access and attacks. Based on the resource-constrained nature of sensor nodes, security mechanisms must be lightweight, energy-efficient, and tailored to the specific characteristics of WSNs. Some of the key aspects of security mechanisms are confidentiality, integrity, authentication, access control, secure key management, physical security, intrusion detection, and secured routing protocols (*Ghadi et al., 2024*; *Urooj et al., 2023*).

To optimize sensor node placement, various deployment strategies and routing protocols are employed. These techniques use specialized algorithms and protocols to strategically position nodes, considering factors like signal coverage, network connectivity, and energy conservation. Routing protocols play a crucial role in energy management, as efficient protocols can minimize power consumption by optimizing data transmission paths and reducing redundant communication.

## OPTIMIZING WSN DEPLOYMENT: STRATEGIES AND ALGORITHMS

In WSN, the deployment of sensor nodes can be executed by considering several performance metrics, such as connectivity, coverage area energy used by the sensor nodes, fault tolerance capacity and security (*Al-Turjman & Lemayian, 2020*) of the data sensed and transmitted. The role of deployment strategies is to deploy sensor nodes that support

high coverage areas with maximized connectivity, low power consumption and error free sensing in multiple types (*Liu, 2015*; *Sadeghi Ghahroudi et al., 2023*). Table 2 shows the strategies and algorithms for efficient deployment based on several metrics using optimization, clustering, and routing protocols. Optimization algorithms (*Yadav & Sharma, 2023*) like metaheuristic-based (*Gunjan, 2023*; *Han et al., 2020*; *Sharma & Gupta, 2020b*) and heuristic algorithms (*Srivastava & Mishra, 2021*) were used to improve the performance of the WSN. Metaheuristics support multi-objective functions (*Kumar et al., 2012*) compared to heuristic algorithms. Particle Swarm Optimization (PSO) (*Cao, Ni & Yin, 2014*; *Nayak et al., 2023*), Greedy Optimization (*Du, Xiao & Fischione, 2018*), Harmony Based Optimization (*Sharma & Gupta, 2017*), Ant Colony Optimization (ACO) (*Deif & Gadallah, 2017*), Ant Lion Optimization (*Li et al., 2023b*), Grey Wolf Optimization (GWO) (*Kaushik, Indu & Gupta, 2019*; *Makhadmeh et al., 2023*), genetic algorithms (GA) (*Benatia et al., 2017*) and swarm intelligence (*Abualigah, Falcone & Forestiero, 2023*; *Yan & Bian, 2023*) were some of the most commonly used algorithms for node deployment. These optimization algorithms can be applied by improving or modifying certain parameters based on the requirements. Apart from optimization algorithms, clustering algorithms (*Bharany et al., 2023*; *Mohapatra et al., 2020*) were effectively used in the node deployment strategy. Clustering and routing (*Daneshvar & Mazinani, 2023*; *Sahoo et al., 2024*) mainly focus on grouping of nodes and performing the function using cluster heads, which provide better nodes by reducing dead nodes and avoiding sensor holes in the coverage area with effective data passing.

Algorithms were formed for deploying a heterogeneous linear WSN based on criteria such as minimizing network delay and balancing power consumption (*Fedorenko et al., 2023*). An energy-efficient automatic adjustment method, *i.e.*, probabilistic coverage virtual force algorithm, is proposed for the automatic-adjustment of randomly deployed homogeneous nodes over a squared area to enhance probabilistic location coverage while retaining network connectivity (*Rout, 2023*). A location algorithm for position accuracy and reduced power consumption using PSO has also been proposed. The deep learning based grouping model energy efficient optimization approach using recurrent neural network (*Surenther, Sridhar & Roberts, 2023*) effective data transmission is proposed with multi-facets.

WSN positioning technology provides real benefits by reducing network configuration errors (*Liang, Wang & Ji, 2022*). The use of multiple constraints aims to save transmission costs by using a minimum number of relay nodes between competitors, extending the lifespan of the network by reducing the maximum effort of relay nodes, and meeting reliability requirements and communication (*Yao et al., 2021*). The sensor placement task is presented as a combinatorial optimization problem and an effective sensor placement paradigm is demonstrated where the stochasticity and dynamics of the environment are observed using conjugate learning automata (*Di, Li & Li, 2020*). Underwater WSNs use a Chemical Reaction Optimization for efficient node placement (*Zhang, Liu & Bi, 2023*), which is a hybrid approach combining the advancements of GA, simulated annealing and ACO. The challenges and limitations associated with large-scale deployment of WSNs include mobile node localization, node heterogeneity, and the varying importance of

**Table 2 Deployment strategies for sensor nodes considering several performances.**

| Reference | Problem | Objective | Algorithm | Optimization | Clustering | Routing |
|---|---|---|---|---|---|---|
| *Liu (2015)* | Coverage, grid-based deployment | Single | Ant colony optimization | ✓ | | |
| *Tarnaris et al. (2020)* | Coverage, k-coverage | Multi | Genetic, Particle swarm | ✓ | | |
| *Zhong & Wang (2018)* | Energy harvesting | Single | Discrete steepest ascent | ✓ | | |
| *Lan (2023)* | IoT fault tolerance | Single | Deep learning, Cluster aware routing | ✓ | ✓ | ✓ |
| *Gola (2024)* | Localization, routing, fault tolerance | Multi | Entropy-based Ant colony optimization | ✓ | ✓ | ✓ |
| *Faris et al. (2023)* | Security issues | Single | Machine learning algorithms | ✓ | ✓ | ✓ |
| *Adu-Manu et al. (2022)* | Energy efficiency, security | Multi | Routing protocol, Network operation protocol | | ✓ | ✓ |
| *Ghadi et al. (2024)* | Security | Single | Machine learning methods | ✓ | | ✓ |
| *Urooj et al. (2023)* | Security | Single | Asymmetric Elliptic Curve Cryptography | | ✓ | ✓ |
| *Sadeghi Ghahroudi et al. (2023)* | Coverage, Connectivity | Single | Force based and Geometric deployment | ✓ | | |
| *Yadav & Sharma (2023)* | Localization | Single | Machine learning and optimization | ✓ | | |
| *Gunjan (2023)* | Lifetime, latency | Multi | Nature inspired metaheuristic based multi-objective optimization | ✓ | | |
| *Sharma & Gupta (2020b)* | Localization | Single | Meta-heuristic | ✓ | | |
| *Han et al. (2020)* | Lifetime | Single | Harmony search, Artificial bee colony | ✓ | ✓ | |
| *Kumar et al. (2012)* | Localization | Single | H-Best particle swarm, Biogeography optimization | ✓ | | |
| *Li et al. (2023b)* | QoS, Coverage, Lifetime | Multi | Ant lion optimizer | ✓ | | |
| *Abualigah, Falcone & Forestiero (2023)* | IoT Challenges | Multi | Swarm intelligence | ✓ | ✓ | ✓ |
| *Yan & Bian (2023)* | Coverage | Single | Sparrow search | ✓ | | |
| *Mohapatra et al. (2020)* | Coverage, lifetime, fault tolerant, Energy | Multi | Clustering approach | | ✓ | |
| *Daneshvar & Mazinani (2023)* | Lifetime, Energy | Multi | Salp swarm Algorithm | ✓ | ✓ | ✓ |
| *Sahoo et al. (2024)* | Lifetime | Single | Intelligent clustering | | ✓ | |
| *Rout (2023)* | Coverage, Connectivity | Single | Probabilistic coverage virtual force algorithm | ✓ | ✓ | |
| *Liang, Wang & Ji (2022)* | Localization | Single | Particle Swarm | ✓ | | |
| *Di, Li & Li (2020)* | Energy efficiency | Single | Conjugate learning automata | ✓ | | |
| *Elfouly et al. (2021)* | Coverage, lifetime | Multi | Heuristic-swarm intelligence | ✓ | | ✓ |
| *Usman et al. (2020)* | Energy efficiency | Single | Adaptive clustering habit | | ✓ | ✓ |
| *Dutta et al. (2023)* | Lifespan | single | Nature inspired optimization | ✓ | ✓ | ✓ |

monitored field subareas. These challenges are addressed through the use of swarm intelligence for efficient deployment (*Elfouly et al., 2021*).

## Strategies for maximizing coverage and connectivity

Network coverage is the key parameter in developing the WSN. Deployment of WSN requires the RoI for sensors to be deployed and monitored. Better coverage of the RoI

should be maintained for better performance of the network (*Dutta et al., 2023*; *Tripathi et al., 2018*). Several coverage issues (*Ghosh, Das & Computing, 2008*; *Zhu et al., 2012*) need to be addressed while deploying the sensor nodes using deployment strategies. Forming obstacle aware network (*Banoth, Donta & Amgoth, 2023*) is a major consideration for enhancing coverage. Algorithms and techniques were developed for better coverage and connectivity of the sensor nodes. These strategies were formed by using static and dynamic sensors where static sensor coverage senses the information without any movement. It tries to cover sensing the whole targeted area. Dynamic sensor coverage is a movable node where the targeted area can be monitored by using unmanned ariel vehicles, aircrafts, *etc.*, It will be helpful to monitor the uncovered area due to the movement of sensors (*Liu et al., 2005*). Maximizing coverage and connectivity (*Al-Karaki & Gawanmeh, 2017*; *Boukerche & Sun, 2018*; *Farsi et al., 2019*; *Wang et al., 2014*) by previous research surveys helps to propose new algorithms for improving performance. Various advanced existing algorithms for maximizing the coverage *i.e.*, Max $\sum_{i=1}^{N} C_i$, where $C_i$ is the coverage performance of i[th] sensor node, and improving connectivity from recent research works were reviewed and explained a few of the updated algorithms for references in the below sections.

### Mutant grey wolf optimizer algorithm (Mu-GWO)

The GWO algorithm is an optimization algorithm generally used for improving coverage in the deployment of sensor nodes. *Nematzadeh et al. (2023)* formed a method by improving this algorithm into a mutant GWO. Mutant-Grey Wolf Optimizer (Mu-GWO) is an improved version of the GWO algorithm, specifically tailored for optimizing the deployment of nodes in WSNs. The original GWO algorithm is a metaheuristic technique inspired by the hunting behavior of grey wolves. Mu-GWO builds upon the principles of GWO but introduces improvements to enhance its efficiency and effectiveness in optimizing sensor node deployment for coverage improvement. leverages the principles of the original GWO algorithm and introduces enhancements tailored specifically for optimizing the deployment of nodes in WSNs. By dynamically selecting the best agents, mutating them to explore new search regions, and optimizing deployment based on predefined criteria, Mu-GWO offers improved efficiency and effectiveness in enhancing sensor node coverage while minimizing deployment costs. Figure 9 shows the working process of Mu-GWO, where alpha is chosen as the constant agent for one of the iterations, and it varies for other agents for further iterations to find the optimal location.

### Steepest descent algorithm with Armijo and Wolf rules (SD-AWR)

*Sheikh-Hosseini & Hashemi (2022)* developed an updated algorithm by improving the direction search method and line search method. The direction search method finds the direction for the node, which is known as a descent direction. Steepest descent (SD) had been used to find the optimal point of direction. Armijo and Wolf rules (AWR) were an iterative optimization approach formed based on line search methods using search directions and step length. This method works using increments calculated by the SD

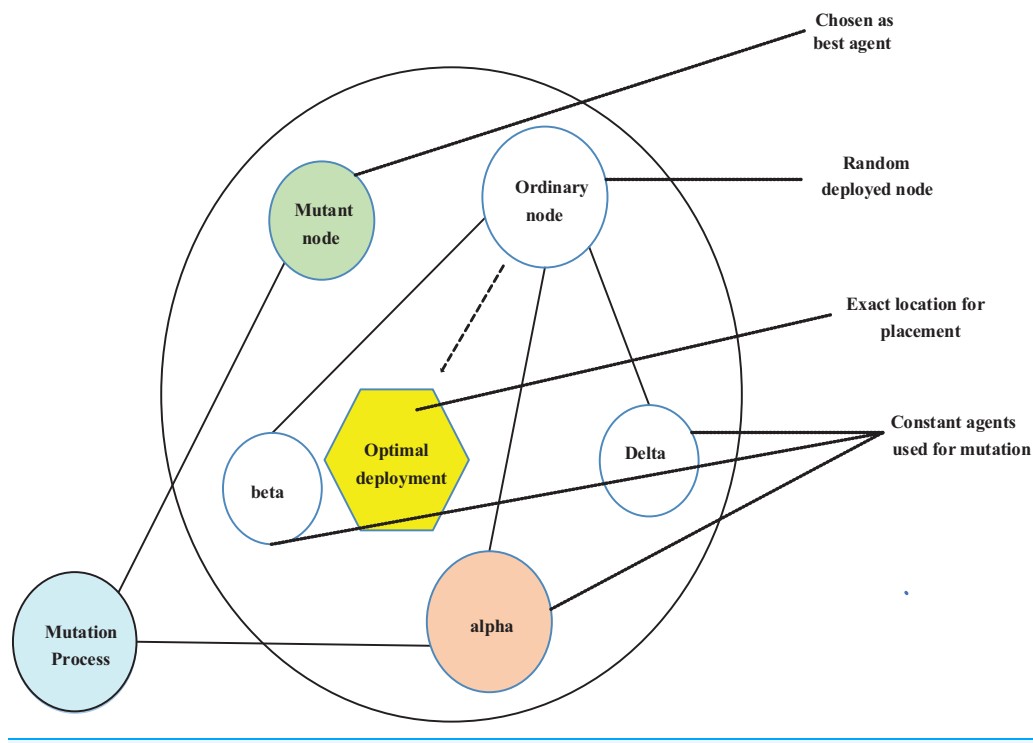

**Figure 9 Mechanism of Mu-GWO.**

algorithm with Armijo and Wolf rules and transmits them to the sensors. Based on the increment, sensors get updated and move to the new location. This process is iterated until the sensor coverage radii are covered. This method enhances the direction search and line search methods for optimizing sensor node deployment in WSNs. By iteratively updating sensor positions based on optimal directions and step lengths, the algorithm aims to improve coverage efficiency in both homogeneous and heterogeneous sensor networks, even in environments with obstacles.

### Harmony search algorithm (HSA)

*Al-Fuhaidi et al. (2020)* identified the deployment problems and found some solutions using the harmony search algorithms using the PSM. This algorithm concentrates on both coverage and cost for the heterogeneous sensor network. Harmony search algorithm (HSA) helps to sense the presence of sensor holes in the area of coverage to solve the hole problem and provide optimal positions for the sensors. HSA provides improvisations in each harmony, choosing the most desirable area. This was done using a few parameters, such as harmonic memory size (HMS), harmonic memory consideration rate (HMCR), random solution rate (RSR), pitch adjustment rate (PAR), and number of iterations (NI). These parameters, after formulating, were applied to PSM for heterogeneous deployment of the sensor nodes shown in Fig. 10 to maximize coverage with reduced nodes, which helps in cost reduction. Thus, HSA with the PSM offers an effective approach to address deployment problems in heterogeneous sensor networks. By optimizing sensor node positions based on coverage and cost considerations, while also detecting and mitigating

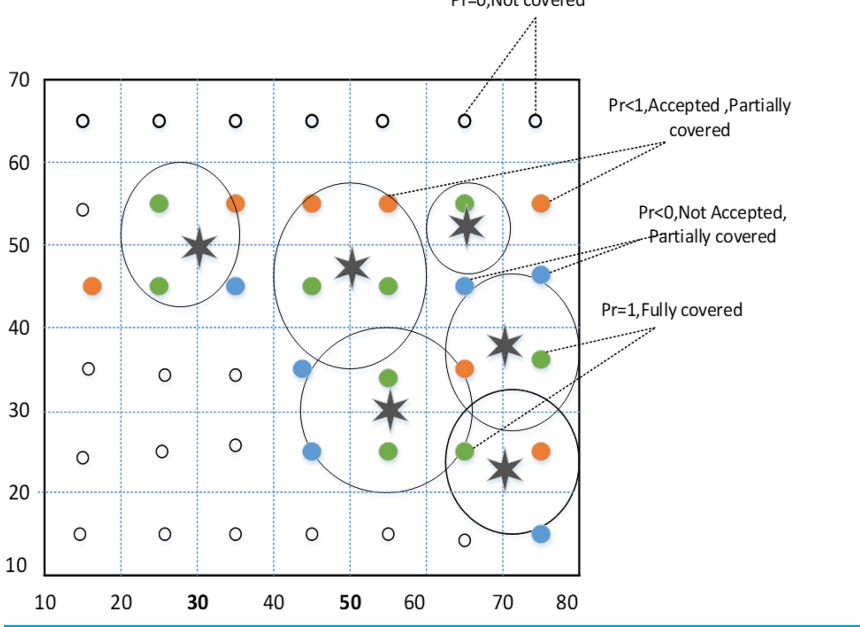

**Figure 10 PSM with harmony search algorithm.**

sensor holes, the algorithm helps achieve efficient and cost-effective sensor deployment for monitoring applications.

### Genetic algorithm for area coverage maximization (GAFACM)

*Tossa et al. (2022)* use the modernized genetic algorithm to address the coverage and connectivity problems. This work is defined by a mathematical model and uses complex objective functions to formulate. This algorithm uses the chromosomes noted as assets of genes, which are applied to solve the problems. Each chromosome had been provided with a fitness function to analyze its quality. Three steps of the formulation process had been done, such as the selection process, the crossover process, and the mutation process. These processes are done for a several iterations to get the fitness values, which can be used to change the location of sensor nodes. This existing work improves coverage and maximizes connectivity. Figure 11 shows the maximized coverage after applying the modified genetic algorithm, showing (A) not being completely covered before optimization and (B), which covers most areas. Thus, the modernized genetic algorithm proposed by *Tossa et al. (2022)* offers a systematic approach by formulating complex objective functions, representing solutions using chromosomes and genes, and iteratively optimizing sensor node placements through selection, crossover, and mutation processes, the algorithm aims to achieve improved coverage and connectivity in WSN deployment.

### Improved greedy strategy search mechanism (IGSSM)

*Yi et al. (2023)* discovered and formulated an improvised greedy search algorithm that is more suitable for underwater sensor networks. Use the depth-first search to check the cover deep into the water. This improved greedy strategy search mechanism (IGSSM) uses

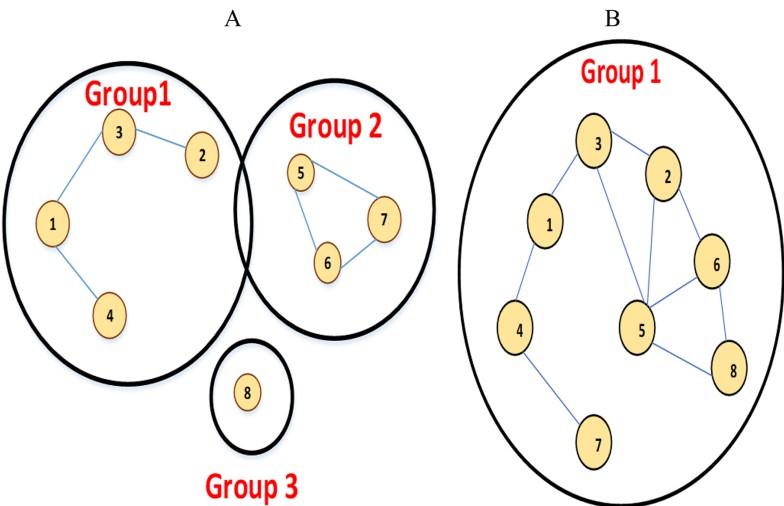

**Figure 11 Genetic algorithm for maximizing coverage.**

grid points to monitor the coverage of the target area at each step. Nodes placed in the grid points cover more target area. This strategy is given by

$$G_i = \arg\max(C_a(g_s)) \tag{3}$$

    $C_a$ – target Coverage area
    $G_s$ – grid points within communication range S
    If this process of depth-first search is replaced by distance-based search algorithms, that would be applicable for target areas other than underwater, *i.e.*, terrestrial networks. *Yi et al. (2023)* leverage depth-first search and grid-based monitoring techniques to optimize coverage and efficiency in underwater sensor networks. By selecting grid points strategically and exploring the underwater environment systematically, the algorithm aims to enhance coverage and monitoring capabilities

### Cooperative particle swarm optimization (CPSO)

*Yarinezhad & Hashemi (2023)* researched and provided a solution for the deployment problem to improve connectivity with a better lifetime using the PSO algorithm with its improvisation. The cooperative PSO technique, followed by improved cooperative PSO using the fuzzy technique, was developed. In cooperative PSO (CPSO), each particle in an individual swarm has one dimension, whereas in standard PSO, it uses n-dimensions for groups of swarms. It was applied to the target area, and the following steps were taken: initialize and encode the particle, form the fitness function, velocity, and position update. This above-explained formulation is suitable for static target areas and solves the K-coverage and Q-coverage problems. Coverage of sensor nodes using random deployment, PSO, and CPSO algorithms in a WSN is shown in Fig. 12. This research offers a novel approach to sensor node deployment optimization in WSNs using PSO and its cooperative variant, augmented with fuzzy techniques. By formulating and solving coverage and connectivity problems, their approach aims to improve network performance and extend network lifetime in static target areas.

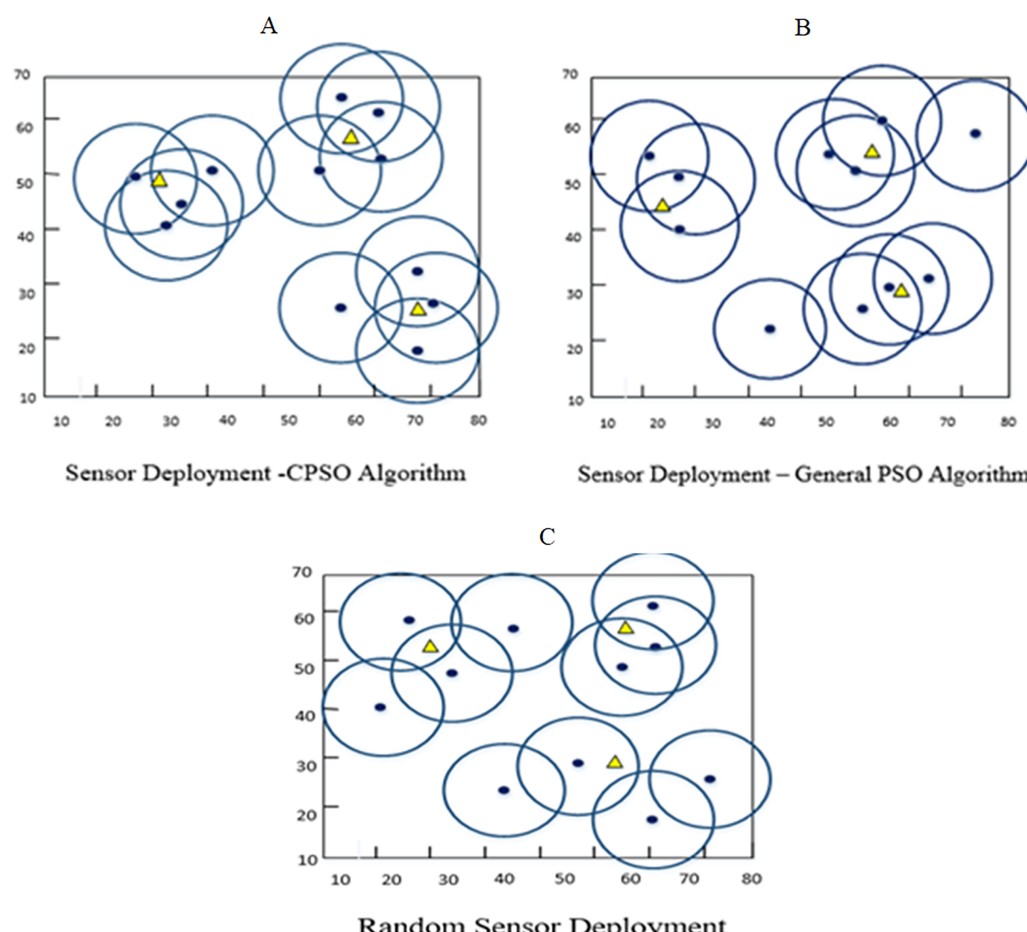

**Figure 12 Sensor node deployment changes for random (C), PSO (B) and CPSO (A) algorithm.**

**Table 3 Deployment strategy comparison for maximizing coverage and connectivity.**

| Reference | Problem | Nodes | Area (m) | Strategy | Highlights | Challenges |
|---|---|---|---|---|---|---|
| Yi et al. (2023) | Depth search | 20 | 50 * 50 | IGSSM | Find targets with improved connectivity | Less lifetime with more power consumption |
| Nematzadeh et al. (2023) | Mutation based | 40 | 100 * 100 | Mu-GWO | Less nodes for coverage | Not suitable for policy-based networks |
| Sheikh-Hosseini & Hashemi (2022) | Optimal location | 60 | 400 * 400 | SD-AWR | Good for homogeneous and heterogeneous sensor. | Extended to Probabilistic nodes for improved lifetime |
| Al-Fuhaidi et al. (2020) | Sensor hole | 50 | 100 * 100 | HSA | Avoids overlapping of Nodes | Not suitable for deterministic deployment |
| Tossa et al. (2022) | Coverage hole | 45 | 100 * 100 | GAFACM | Covers whole target area | Outperforms for connectivity disruption |
| Yarinezhad & Hashemi (2023) | K coverage, Q-coverage | 100 | 500 * 500 | CPSO | Different coverage problems be optimized | Not much suitable for IoT based deployment of sensor nodes |

Table 3 presents a comparative analysis of the algorithms discussed earlier, focusing on their effectiveness in maximizing coverage, along with their respective benefits and potential areas for future improvement. Additionally, the table includes information regarding the number of nodes utilized in the analysis and the simulated area size for each algorithm.

## Strategies for optimizing power usage

An increase in the necessity of sensor networks requires a greater number of deployments of nodes that are localized to have a better network lifetime. The power utilization of sensor nodes in WSNs is a critical aspect that directly impacts longevity, overall efficiency (*Kori et al., 2009*), and network performance given by $\text{Min} \sum_{i=1}^{N} E_i$, where $E_i$ is the energy consumption of i$^{\text{th}}$ sensor node. Various strategies, like routing algorithms were found by researching multiple research articles for the energy-efficient deployment (*Rault, Bouabdallah & Challal, 2014*; *Usman et al., 2020*) of sensor nodes for various WSNs with reduced power consumption and for a long-lasting life of sensor nodes. Metaheuristic-based strategies (*Sharma & Gupta, 2020a*) for energy efficiency provide better optimization. A node deployment for ad-hoc sensor networks considering energy optimization with uncertain sensing addressed using routing aware Lloyd optimization algorithm (*Guo, Karimi-Bidhendi & Jafarkhani, 2020*).

### Energy-balanced neuro fuzzy dynamic clustering

*Chithaluru et al. (2023)* developed an algorithm for IoT-based devices for efficient resource management using energy-efficient dynamic clustering routing (EEDCR). By using neural networks, the dynamic clusters were formed. Cluster head selection was done using neural fuzzy logic. This logic also helps in achieving less packet delay and network overhead, which helps in resource management with a better lifetime. A gradient-based neural network, shown in Fig. 13, is used, consisting of a sensing unit and a competitive layer to communicate with the neurons used in neuro-fuzzy logic. The above-explained algorithm provides better simulation results compared with LEACH (*Daanoune, Abdennaceur & Ballouk, 2021*) and LEACH-C (*Khediri et al., 2014*) algorithms. This research presents an innovative approach to resource management in IoT-based devices, utilizing EEDCR coupled with neural networks and fuzzy logic. This approach addresses key challenges in IoT networks such as energy consumption, packet delay, and network overhead, leading to improved performance and longevity of the network compared to traditional routing algorithms.

### Threshold enabled scale and energy efficient scheme (TESEES)

*Abdul-Qawy et al. (2023)* stated a scheme for improving energy savings in heterogeneous WSN. The algorithm threshold enabled scalable and energy efficient scheme (TESEES) supports avoiding frequent unnecessary data transmission and reducing energy dissipation. This algorithm uses the zone formation technique as shown in Fig. 14, which

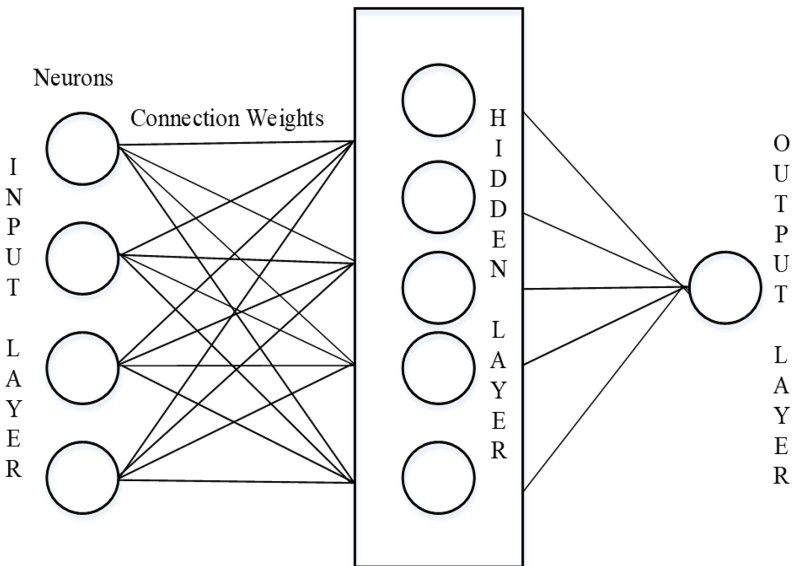

**Figure 13 Feed forward neural network.**

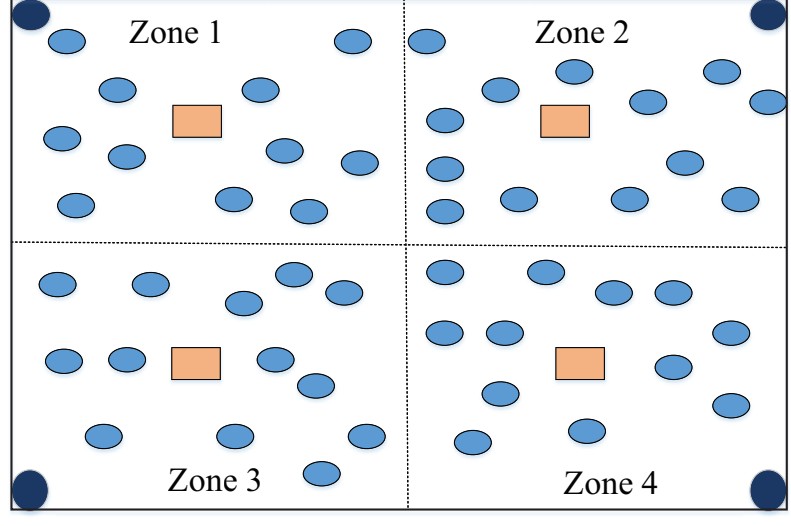

**Figure 14 Zone formation in coverage area.**

uses zone aggregator (ZA) nodes in each formed zone. Use of heterogeneous nodes and energy harvesting relay (EHR) nodes for the efficient routing of data. After the formation of zones with different nodes, event thresholding is performed. Using the grid of EHR nodes helps with energy savings. TESEES uses threshold-based minimum cost cross-layer

transmission (TMCCT), which selects the data transmission route after sensing and sends data to base station nodes. TESEES offers a holistic approach to prolonging WSN lifetime and conserving energy in resource-constrained environments.

### Greedy-like step-by-step trimming heuristic algorithms (GLSTA)

Zhao et al. (2023) researched and proposed an algorithm for WSN network lifetime improvement with energy efficiency. The algorithm named greedy-like step-by-step trimming heuristic algorithm (GLSTA), removes unnecessary nodes by reducing the redundancy, which helps in energy savings. This method uses the organization cluster head and data cluster head for the proper functioning of the cluster during the movement of the target. This algorithm is a step-by-step process that includes four processes: initialization of nodes, cluster formation, node scheduling, and data transmission. The suggested algorithm is an iterative process repeated n times to get effective results. As a result, this technique is an optimization method that is primarily employed in extensively deployed WSNs to schedule the appropriate jobs to cluster. Overall, GLSTA offers an optimization method for extensively deployed WSNs, focusing on scheduling appropriate tasks and removing unnecessary nodes to enhance energy efficiency and prolong the network lifetime. Through its step-by-step trimming heuristic approach and cluster organization, GLSTA provides a systematic framework for improving WSN performance in energy-constrained environments.

### Combing clustering and energy aware routing protocols (CCEAR)

Jannu et al. (2023) suggested a model for managing energy in IoT networks. The algorithm is formed by combining clustering mechanisms and energy-aware routing protocols (Pantazis, Nikolidakis & Vergados, 2012), which are used to reduce hotspot problems that have energy holes. This combined algorithm also helps in network lifetime improvement using energy harvesting nodes. Clustering of nodes and routing for data transmission, shown in Fig. 15, are the two main phases analyzed to perform the deployment using the algorithm mentioned above. The first phase includes cluster formation and the assignment of cluster heads (Singh et al., 2023) in each cluster later phase includes next-hop selection and data routing. If a cluster head (CH) fails to transmit data, the responsibility of data transmission falls upon an energy harvesting (EH) node, which acts as the next hop in the communication path. This approach ensures robustness and reliability in data transmission, even in the event of CH failures.

### Voronoi-glow-worm swarm optimization K-means algorithm (VGWSO)

Chowdhury & De (2021) proposed an algorithm for the proper coverage of sensors and better energy efficiency during deployment. It is a multi-objective optimization algorithm that addresses both coverage and energy problems to enhance network lifetime. Voronoi Glow-worm Swarm Optimization K-means algorithm uses the Voronoi cell placement structure shown in Fig. 16 along with the glow-worm swarm and the K-means optimization algorithm (El Khediri et al., 2020) to minimize active nodes with high coverage. At first, the nodes use the K-means algorithm to form clusters, then, using the Voronoi structure (Ammarfaizal, Putrada & Abdurohman, 2021; Shu et al., 2019) the

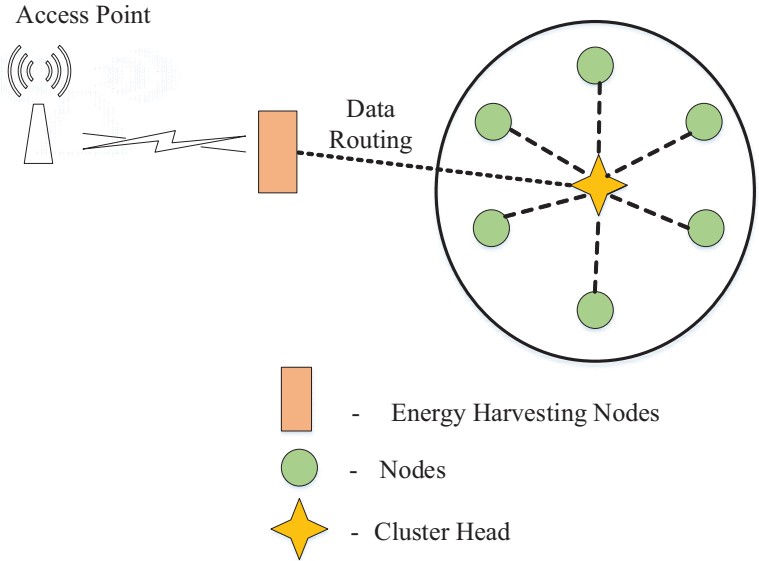

**Figure 15 Data transmission using energy harvesting relay nodes.**

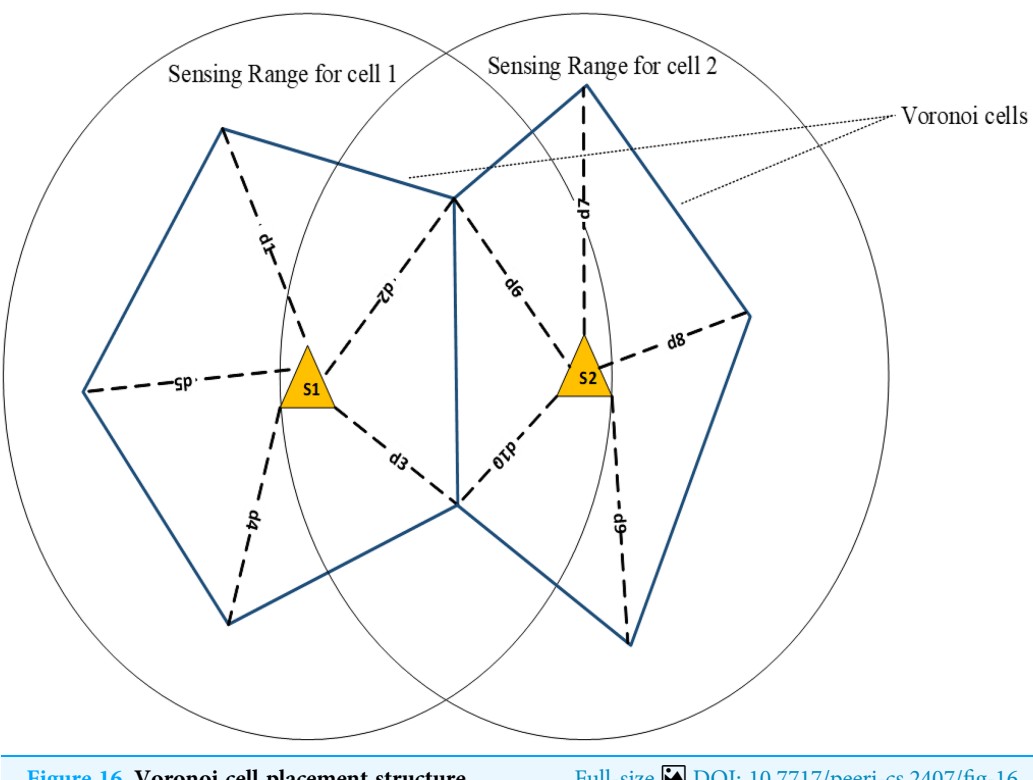

**Figure 16 Voronoi cell placement structure.**

coverage radius is calculated, and glow-worm swarm optimization (GSO) helps find the optimal position for sensors. The use of multi-hop transmission (*Choi & Lee, 2021*) along with the sleep wake mechanism reduces energy consumption. By leveraging Voronoi GSO, K-means clustering, multi-hop transmission, and sleep-wake mechanisms, the algorithm

aims to optimize sensor deployment and prolong the network lifetime. However, its effectiveness may vary depending on the environmental conditions, particularly in obstacle-rich environments where communication and coverage challenges are more pronounced.

### Improved memetic algorithm and node cooperation strategy (IMA-NCS)

*de Brito et al. (2022)* proposed a scheduling mechanism that supports an energy-efficient network with a longer life. The memetic algorithm (*Dowlatshahi, Rafsanjani & Gupta, 2021*) is an advancement of genetic algorithms with more search operations in it. Formation and use of multi-objective fitness functions, *i.e.*, genetic codes, find the working nodes in the network. This improved memetic algorithm (IMA) results in effective node scheduling. This algorithm supports three objectives, which are formulated and expressed in the below equations.

i) Reducing working nodes

$$min\ m_1 = \frac{1}{N}\sum_{I=1}^{N}\varepsilon_i \tag{4}$$

ii) Improving coverage and connectivity

$$max\ m_2 = \frac{1}{N}\sum_{I=1}^{N}\varepsilon_i \times \delta(s_i) \tag{5}$$

$\delta(s_i)$ gives the relationship between the transmission power of nodes, receive reliability of nodes, and path loss.

iii) Improving energy efficiency

$$max\ m_3 = \frac{1}{N}\sum_{I=1}^{N}\frac{E_f - E_{tot}}{E_i} \tag{6}$$

The node cooperation strategy shown in Fig. 17 was formed to avoid unnecessary data retransmission, which leads to more energy consumption. The node cooperation strategy (NCS) supports and reduces unwanted traffic load with improved network lifetime by using effective data transmission.

Table 4 showcases a comparative analysis of the algorithms discussed earlier, aiming to enhance energy efficiency. It delineates their respective advantages, challenges, and potential avenues for future development. Furthermore, the table provides specifics regarding the number of nodes employed in the analysis and the simulated area dimensions for each algorithm.

## COMPARISON AND RESULTS

The application of sensors is vast in the development of new technologies as well as in ensuring secure data monitoring. WSN, by means of communicating and processing the real-time world, has played a better role in change and progress in this modern era. Proper deployment of nodes sensing and processing the needed information for the application to

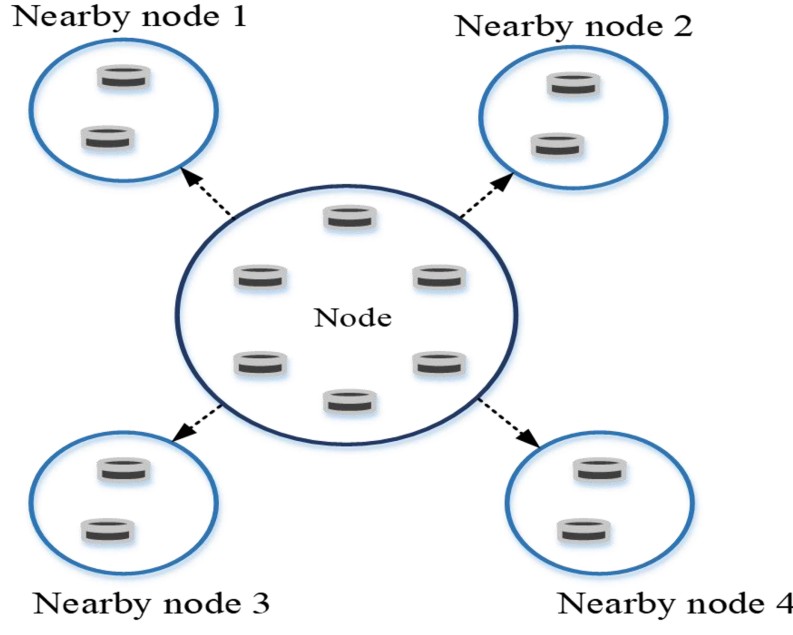

**Figure 17** Node cooperation strategy.

**Table 4 Deployment strategy comparison for improving Energy saving.**

| Reference | Problem | Nodes | Area (m) | Strategy | Highlights | Challenges |
|---|---|---|---|---|---|---|
| *Zhao et al. (2023)* | Dead nodes | 400 | 300 * 300 | GLSTA | Dual cluster head improve energy consumption | Improved for large scale heterogeneous networks |
| *Abdul-Qawy et al. (2023)* | Energy dissipation | 1,000 | 400 * 400 | TESEES | Avoid unwanted frequent data transmission | Enhance protocol for large scale deployment |
| *Chowdhury & De (2021)* | Energy and lifetime saving | 150 | 100 * 100 | VGWSO | Multi-hop transmission and sleep-wake mechanism improve efficiency | Not suitable for more obstacles in sensing region |
| *Jannu et al. (2023)* | Hotspot problems | 250 | 300 * 300 | CCEAR | Reduced energy holes | Not suitable for coverage area for continuous monitoring |
| *Chithaluru et al. (2023)* | Network overhead | 100 | 100 * 100 | EEDCR | Dynamic clustering improves network lifetime | Restricted resources of IoT devices |
| *de Brito et al. (2022)* | Node Scheduling | 250 | 300 * 300 | IMA-NCS | Balance energy usage of nodes to extend network lifespan | Not suitable for complicated scenarios, improve stability of algorithm |

which they are deployed can help evaluate the network's performance. There are many deployment strategies other than the various algorithms proposed by different researchers aimed at overcoming problems in existing deployment methods. An open-ended issue has to be addressed when analyzing the comparison table of some algorithms that have existed recently. Tables 3 and 4 challenges can be noticed. Most of these algorithms have a single objective, as noted in Table 5. It is important to address both energy and coverage issues in one solution to enhance network performance. The earlier mentioned algorithms were used to test their performances on coverage performance and efficient energy consumption, while Figs. 18 and 19 show variations in their efficiency respectively.

**Table 5 Existing algorithms comparison in terms of connectivity and lifetime under various Parameters.**

| Objective | Reference | Algorithm | Sensor nodes | Space | Deploy type | Objective | Connectivity | Life time |
|---|---|---|---|---|---|---|---|---|
| Coverage maximization | *Yi et al. (2023)* | IGSSM | Homo, Hetero | 3D | Random | Single | ↑ | ↓ |
| | *Yarinezhad & Hashemi (2023)* | CPSO | Hetero | 2D | Random | Single | ↑ | ↔ |
| | *Nematzadeh et al. (2023)* | Mu-GWO | Homo, Hetero | 2D | Random | Single | ↑ | ↔ |
| | *Sheikh-Hosseini & Hashemi (2022)* | SDAWR | Homo, Hetero | 2D | Random | Single | ↔ | ↔ |
| | *Al-Fuhaidi et al. (2020)* | HSA | Hetero | 2D | Random | Multi | ↑ | ↑ |
| | *Tossa et al. (2022)* | GAFACM | Homo | 2D | Random | Single | ↔ | ↓ |
| Energy efficiency maximization | *Chithaluru et al. (2023)* | EEDCR | Homo, Hetero | 2D | Random | Single | ↓ | ↔ |
| | *Jannu et al. (2023)* | CCEAR | Homo, Hetero | 2D | Random | Single | ↔ | ↔ |
| | *Zhao et al. (2023)* | GLSTA | Homo, Hetero | 2D | Random | Single | ↔ | ↔ |
| | *Abdul-Qawy et al. (2023)* | TESEES | Homo, Hetero | 2D | Random | Single | ↔ | ↔ |
| | *Chowdhury & De (2021)* | VGWSO | Hetero | 2D | Random | Multi | ↑ | ↑ |
| | *de Brito et al. (2022)* | IMA-NCS | Hetero | 3D | Random | Multi | ↑ | ↑ |

**Note:**
Symbol representation: ↑: High, ↔: Medium, ↓: Low

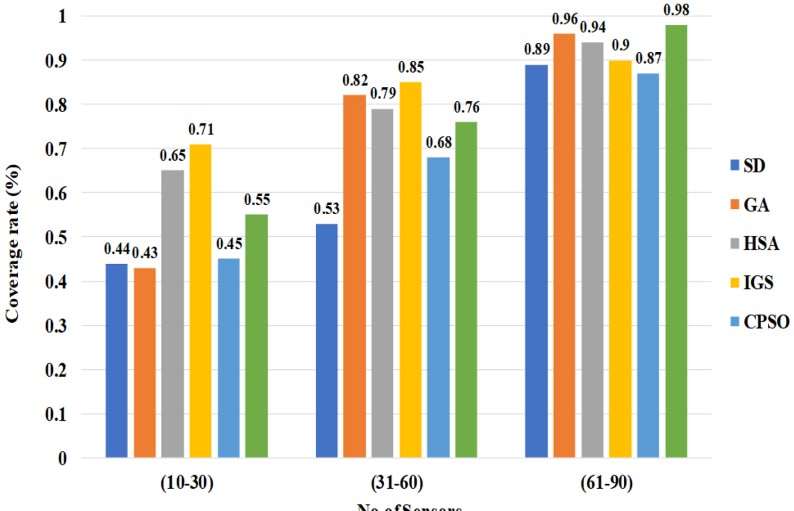

**Figure 18 Variations in coverage efficiency for various algorithm.**

## Summary and key findings

WSNs focus on real-time communications and processing, greatly contributing to modern technological progress. This study emphasizes the importance of properly deploying sensor nodes to effectively sense and process the necessary information, which is vital

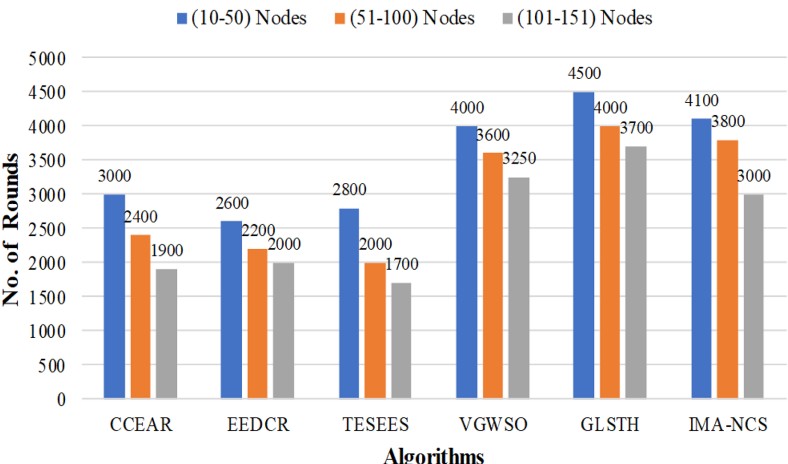

**Figure 19 Variations in energy efficiency for various algorithms.**

for evaluating network performance. It discusses one major input related to the strategies for deploying sensor nodes. There is an essential requirement to deploy the nodes to test the performance of the networks because if the sensor nodes are not properly deployed, they will not sense or process the required information. Previous deployment strategies and algorithms proposed by other researchers have been reviewed in the article, which, are designed to deal with various challenges in the existing methods for deployment.

One of the critical observations made is that most algorithms designed so far are focused on either one objective, namely coverage or energy efficiency. This one-dimensionality is reflected in the analysis of those algorithms in Tables 3–5. It is argued in the article that enhancement in terms of the network means consideration of both energy and coverage issues together. In this case, dual-objective optimization will result in more balanced and efficient WSNs. In this article, a few algorithms are evaluated in terms of their performance concerning coverage and energy efficiency. These different plots, as represented in Figs. 18 and 19, contribute to the efficiency variation of the algorithms and provide valuable insights into their relative performance. According to this study, since challenges have been identified, there are many needs for developing new algorithms or methodologies that can enable the overcoming of such challenges and the new solutions shall be validated and compared with the existing solutions. To conclude, this article addresses the research questions, aimed at improving further studies. The key findings constitute the improvement of deployment strategies, the constraints of the current algorithms, the development of dual-objective optimization methods, and new algorithms that counteract existing challenges. These questions allow researchers to progress the field of WSNs even further, gaining better and more efficient network solutions. Moreover, additional research is required to address the challenges of using multi-objective functions

and to resolve persistent implementation issues in wireless sensor network deployment strategies.

## CHALLENGES AND FUTURE WORK

Some WSN case studies include forest fire detection, air and water quality monitoring, and patient health management by wearing sensors. WSN in agriculture for precision farming and livestock monitoring using real-time data regarding soil conditions and animal health. Applications like optimization of traffic flow, vehicle parking, and energy-efficient street lighting in smart cities. Other industrial applications include machine condition monitoring and inventory management. These applications identify the role WSNs can play in improving efficiency, safety, and decision-making across various sectors. A lot of challenges arise while deploying the sensor nodes in various environments. From the survey of various recent deployment techniques, some drawbacks were found, and future improvements in the field need to be considered for further research.

*Challenge 1: Obstacle constraint algorithms*

Obstacles within the deployment area may not be adequately addressed by current methods, leading to less optimal coverage. Future research could concentrate on developing obstacle constraint-based algorithms that will effectively navigate and deploy sensor nodes around obstacles. To dynamically adapt to changing obstacle scenarios in real-time, alternative techniques such as ML or AI-based algorithms can be used.

*Challenge 2: Multi-Objective Techniques*

The complexities and trade-offs involved in sensor node deployment may not be fully captured by single-objective algorithms. Future work can include making coherent multi-objective optimization algorithms that consider many conflicting objectives together, like coverage, connectivity, energy savings, cost, and lifetime, at once. Evolutionary algorithms, swarm intelligence techniques, or hybrid approaches are worth exploring for the effectiveness of the above challenge.

*Challenge 3: Wireless Rechargeable Sensor Networks (WRSN)*

The use of wireless rechargeable sensor networks can improve the coverage and energy efficiency of sensor node deployment. Further future work could be on optimizing the design and management of WRSNs to enhance coverage, lengthen network lifetimes, and reduce maintenance efforts. The use of WRSNs can improve coverage and energy efficiency during the deployment of sensor nodes. For instance, optimizing the design and management of WRSNs to enhance coverage, prolong network lifetime, and minimize maintenance efforts is an area for future research. Promising research directions include investigating novel charging strategies, energy harvesting techniques, and efficient routing protocols specific to WRSNs.

*Challenge 4: Large-Scale Deployment with Unmanned Aerial Vehicles (UAV)*

When deploying sensors in large areas using UAVs, there are significant challenges that must be faced to optimize deployment efficiency and provide enough coverage. On the other hand, in future work, one could look into the integration of deep learning algorithms

with UAV-based deployment strategies to automate and optimize the process. In addition, reinforcement learning for path planning as well as object detection for obstacle avoidance and predictive modelling for optimal sensor placement can be expanded as means of handling challenges facing large-scale UAV deployment.

## CONCLUSION

The article provides an overview of WSN sensor deployment research done in different settings employed in various applications, such as smart cities. Coverage, connectivity, energy efficiency, and lifetime were given focus while deploying strategies with multiple objectives because they are interdependent. Better network life could be achieved by applying better coverage and the best connection with the best energy-conserving protocols. While developing the deployment strategies, metrics such as deployment type, sensor node type, and coverage area have been considered. From this survey analysis, we can observe the dependence of algorithms on the environment, and each algorithm possesses unique advantages only in a specific environment while others have none. One has to take into consideration the entire area where sensors will be deployed first before selecting an algorithm to use when doing it. The deployment strategy varies depending on the target area and the hindrances that occur over its coverage location in different environments. Using multi-objective functions for deployment results in less complicated strategies. However, there are also a multitude of challenges during deployments, which are very critical and should be considered for further studies. Research work still lies ahead, with numerous counts of issues regarding their implementations so far.

## LIST OF ABBREVIATIONS

| | |
|---|---|
| **WSN** | Wireless Sensor Networks |
| **IoT** | Internet of Things |
| **ML** | Machine learning |
| **DL** | Deep Learning |
| **AI** | Artificial Intelligence |
| **RoI** | Region of Interest |
| **BSM** | Binary Sensing Model |
| **PSM** | Probabilistic Sensing Model |
| **Mu-GWO** | Mutant Grey Wolf Optimization |
| **SD-AWR** | Steepest Descent with Armijo and Wolf Rules |
| **CPSO** | Cooperative Particle Swarm Optimization |
| **IGS** | Improved Greedy Strategy |
| **HSA** | Harmony Search Algorithm |
| **GA** | Genetic Algorithm |
| **EEDCR** | Energy Efficient Dynamic Clustering Routing |
| **CCEAR** | Combined Clustering and Energy Aware Routing |
| **GLSTH** | Greedy-like step-by-step Trimming Heuristic Algorithm |
| **TESEES** | Threshold Enabled Scale and Energy Efficient Scheme |

| VGWSO | Voronoi-Glow worm Swarm Optimization |
| IMA-NCS | Improved Memetic Algorithm and Node Cooperation Strategy |
| WRSN | Wireless Rechargeable Sensor Network |
| UAV | Unmanned Aerial Vehicle |

### Funding
This work was supported by the Korea Environmental Industry & Technology Institute (KEITI), with a grant funded by the Korea Government, Ministry of Environment (the development of IoT-based technology for collecting and managing big data on environmental hazards and health effects), under Grant RE202101551. The funders had no role in study design, data collection and analysis, decision to publish, or preparation of the manuscript.

### Grant Disclosures
The following grant information was disclosed by the authors:
Korea Environmental Industry & Technology Institute (KEITI).
Korea Government, Ministry of Environment: RE202101551.

### Competing Interests
The authors declare that they have no competing interests.

### Author Contributions
- Anusuya P. conceived and designed the experiments, performed the computation work, prepared figures and/or tables, and approved the final draft.
- Vanitha C. N. performed the experiments, analyzed the data, performed the computation work, authored or reviewed drafts of the article, and approved the final draft.
- Jaehyuk Cho performed the computation work, authored or reviewed drafts of the article, and approved the final draft.
- Sathishkumar Veerappampalayam Easwaramoorthy conceived and designed the experiments, performed the experiments, analyzed the data, performed the computation work, prepared figures and/or tables, authored or reviewed drafts of the article, and approved the final draft.

### Data Availability
This is a literature review.

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
