# Peer review of "A comprehensive review of sensor node deployment strategies for maximized coverage and energy efficiency in wireless sensor networks"

_PeerJ Computer Science, doi:10.7717/peerj-cs.2407_

## Round 0.1 · original submission · Major Revisions

This paper is a review article focuses on sensor node deployment strategies for wireless sensor networks. After the first round of review, a "Major Revision" is recommended and authors should also consider the following comments.

1. In the "Abstract", authors should be rewritten this section as it is too generic and it has no technical contents, i.e. equations/formulas in it. Authors should highlight the contributions of this paper as a review article. How this paper is different than other review articles in the literature ?

2. What technical findings of this paper can inform other researchers in the field to further advance their research ?

3. What the clear research questions that authors would like to answer in this paper ?? These are not clear in the paper.

4. The methodology section of this paper should be clarified to explain how this review was carried out.

·

Basic reporting

Clarity and Language:
The manuscript is written in clear and professional English. However, there are several instances where the language can be improved for better clarity and readability.

For example, (Lines 53-74): The introduction is comprehensive, but some sentences are overly complex. For instance, "WSNs consists of interconnected sensors that communicate data wirelessly, either to a central system or to other sensors" could be simplified. Consider breaking down complex sentences and ensuring each sentence conveys a single clear idea.

Consider rephrasing "WSNs is categorized into two main groups: centralized and distributed" to "WSNs are categorized into two main groups: centralized and distributed."

Structure and Context:

The structure of the manuscript conforms to PeerJ standards and includes all necessary sections such as Abstract, Introduction, Survey Methodology, General Concepts, and others.

The Introduction provides a good context for the study, explaining the importance of WSNs and the challenges in sensor node deployment.

The literature is well-referenced and relevant, providing a comprehensive overview of the field.

Literature Review and Referencing:

The Introduction provides a good context for the study, explaining the importance of WSNs and the challenges in sensor node deployment.

The literature is well-referenced and relevant, providing a comprehensive overview of the field.

Experimental design

The survey methodology is well described, with clear research questions guiding the investigation.

Ensure that the methodology section includes enough detail for replication. For example, the process of selecting and reviewing articles should be more elaborately described.

Validity of the findings

The review covers a wide range of topics related to WSN deployment, including optimization algorithms, deployment strategies, and performance evaluations.

The manuscript successfully identifies recent advancements and potential areas for further research.

The conclusions are well-stated and linked to the original research questions.

Additional comments

Suggestions for Improvement:

Language and Grammar: Needs to be reviewed

Detailed Methodology: Expand on the survey methodology to provide a clearer picture of how the literature review was conducted.

Future Directions: Elaborate on the future research directions and unresolved questions identified in the conclusions.

Real-World Examples: Incorporate more real-world examples or case studies to illustrate the practical applications and challenges of WSN deployments.

·

Basic reporting

Clear, unambiguous, professional English language is used throughout the paper. The introduction and background provide sufficient context for readers. The review of literature is well referenced and revelant. The structure of the paper conforms to Peer J standards; it is within the norms of the discipline and is clear. The topic and study presented within the paper are within the scope of the journal. However, it is not clear how much cross-disciplinary appeal the paper will have. It would be improved by specifically identifying several other disciplines that would benefit from the presented study on wireless sensor networks.

There are at least three well cited reviews related to sensor node deployment in wireless sensor networks that have been published since 2020. They are highlighted below. Each has been cited at least 175 times. The first and third are referenced by the author but only as sources for what wirtelss networks and deployment strategies are; this study is not distinguished from either of them. The authors need to motivate the reason and novelty for this review for the audience.

Priyadarshi, Rahul, Bharat Gupta, and Amulya Anurag. "Deployment techniques in wireless sensor networks: a survey, classification, challenges, and future research issues." The Journal of Supercomputing 76 (2020): 7333-7373.

Kandris, Dionisis, et al. "Applications of wireless sensor networks: an up-to-date survey." Applied system innovation 3.1 (2020): 14.

BenSaleh, Mohammed Sulaiman, et al. "Wireless sensor network design methodologies: A survey." Journal of Sensors 2020.1 (2020): 9592836.

Experimental design

The article content is within the aims and scope of the journal. The investigation is done to an ethical standard. However, the technical standard to which it is performed could be improved by identifying how artciles for inclusion in the review were identified. Was an objective, repeatable, criteria applied as is typically done in a systematic review. Were multiple raters employed to identify which papers were relevant to which research questions. The paper lacks information on the methodology of the literature review; making the presented results impossible to trust. As a result, The methods are not described with sufficient detail & information to replicate. Similarly, the lack of details on the methodology make it impossible to determine if it is comprehensive or unbiased.

The souces included are adequatedly cited, quoted, and paraphrased. However, the organization of the review could be improved. Spending more time discussing the methodology and providing a better roadmap for readers and reviewers about how the results will be presented to them before just going through all of them would improve things.

Validity of the findings

Due to the lack of the methodology meaningful replication of the study is not possible. Furthermore, the rationale and benefit of this study (compared directly to other related recent studies) needs to be more clearly stated in the paper. The conclusions of the review are well stated and linked to seven originial research questions. There is a developed and supported argument that meets the goals set out in the introduction; however, the soundness of that argument is questionable without additional details on the review methodology used by the authors. The conclusion provides answers to the research question and several new avenues for future work.

Additional comments

It is unclear how novel the provided review is given several established recent reviews of related work. In addition, the lack of methodology provided about how papers were chosen and analyzed to answer research questions, specifically, if such a process was objective and repeatable by others is a major weakness of this manuscript.

·

Basic reporting

Authors did nice work on WSN survey.
However Need to concentrate on the following
1. The images are not in good quality.
2. Lot of papers are missing with respect energy efficiency. Need to do thorough research of high quality paper, lots of papers are missing.
3. There are other parameters to consider in addition to Energy efficiency.

Experimental design

1. Title is not matching with work presented in the paper.
2. Lot more good papers are missing for node deployment
3. Abstract is deviating from the paper.

Validity of the findings

no comment

---

## Round 0.2 · accepted · Accept

Thank you for revising the paper in response to the reviewers' comments. As all the journal's concerns have been suitably addressed, I recommend that the paper be accepted in its current form.

·

Basic reporting

The basic reporting concerns of the paper have been addressed to my satisfaction. The paper now passes all basic reporting criteria.

Experimental design

The study design concerns of the paper have been addressed to my satisfaction. The paper now passes all study design criteria.

Validity of the findings

The validity of the findings concerns of the paper have been addressed to my satisfaction. The paper now passes all validity of the findings criteria.

Additional comments

My concerns have been sufficiently addressed. The paper is now suitable for publication.

·

Basic reporting

• Clear, unambiguous, and professional English is used throughout the paper.
• Sufficient information in the introduction for a clear understanding of WSN is provided and the literature is recent and well referenced.
• The structure of this article is perfectly conforming to PeerJ standards and well improved for clarity.
• Yes, the review is broad by including multiple topics of WSN which is within the scope of the journal but cross-disciplinary other than WSN may be considered.

Experimental design

• The article is within Scope of the journal and the investigation performed is more technical and ethical.
• The survey methodology explained with the figures is found to be unbiased.

Validity of the findings

• The Survey gives new findings and updates in WSN deployment shows its novelty.
• Conclusions are well stated with the performance comparison of recent algorithms
and connected with future gaps to resolve.

Additional comments

• This article contributes to the current issues in WSN node deployment for maximized coverage and energy efficiency.
• This paper helps to improve the existing algorithms for real-time applications like smart cities, and environmental monitoring.

Reviewer 5 ·

Basic reporting

Can be Improved.
After reading the whole paper, following observations have been made.
1. The usage of the English language should be improved in the paper.
2. The abstract section needs improvement.
3. The technical depth of the paper is limited and should be improved.
4. Reference section needs improvements.

Experimental design

.

Validity of the findings

.

Additional comments

.